# High-Density Lipoprotein Subclasses and Their Role in the Prevention and Treatment of Cardiovascular Disease: A Narrative Review

**DOI:** 10.3390/ijms25147856

**Published:** 2024-07-18

**Authors:** Qiaofei Chen, Ayiguli Abudukeremu, Kaiwen Li, Minglong Zheng, Hongwei Li, Tongsheng Huang, Canxia Huang, Kexin Wen, Yue Wang, Yuling Zhang

**Affiliations:** 1Department of Cardiology, Sun Yat-sen Memorial Hospital, Sun Yat-sen University, Guangzhou 510120, China; chenqf56@mail2.sysu.edu.cn (Q.C.); ayglabdkrm@163.com (A.A.); zhengmlong3@mail2.sysu.edu.cn (M.Z.); lihw57@mail2.sysu.edu.cn (H.L.); huangtsh3@mail2.sysu.edu.cn (T.H.); huangcx5@mail.sysu.edu.cn (C.H.); wenkx@mail2.sysu.edu.cn (K.W.); wangy2323@mail2.sysu.edu.cn (Y.W.); 2Guangdong Provincial Key Laboratory of Malignant Tumor Epigenetics and Gene Regulation, Guangdong-Hong Kong Joint Laboratory for RNA Medicine, Medical Research Center, Sun Yat-sen Memorial Hospital, Sun Yat-sen University, Guangzhou 510120, China; 3Nanhai Translational Innovation Center of Precision Immunology, Sun Yat-sen Memorial Hospital, Foshan 528200, China; 4Zhongshan School of Medicine, Sun Yat-sen University, Guangzhou 510120, China; likw9@mail2.sysu.edu.cn; 5Guangdong Province Key Laboratory of Arrhythmia and Electrophysiology, Guangzhou 510080, China

**Keywords:** HDL subclasses, cardiovascular disease, cardiovascular risk, nomenclatures, recombinant HDL

## Abstract

The association between high-density lipoprotein cholesterol (HDL-C) and cardiovascular disease (CVD) is controversial. HDL-C is one content type of high-density lipoprotein (HDL). HDL consists of diverse proteins and lipids and can be classified into different subclasses based on size, shape, charge, and density, and can change dynamically in disease states. Therefore, HDL-C levels alone cannot represent HDLs’ cardioprotective role. In this review, we summarized the methods for separating HDL subclasses, the studies on the association between HDL subclasses and cardiovascular risk (CVR), and the impact of lipid-modifying medications and nonpharmacological approaches (exercise training, dietary omega fatty acids, and low-density lipoprotein apheresis) on HDL subclasses. As HDL is a natural nanoplatform, recombinant HDLs (rHDLs) have been used as a delivery system in vivo by loading small interfering RNA, drugs, contrast agents, etc. Therefore, we further reviewed the HDL subclasses used in rHDLs and their advantages and disadvantages. This review would provide recommendations and guidance for future studies on HDL subclasses’ cardioprotective roles.

## 1. Introduction

Cardiovascular disease (CVD) remains the leading cause of death globally, the number of people who died from CVD in 2019 was 18.6 million, and it is projected to increase to more than 23.6 million by 2030 [1,2]. High-density lipoprotein (HDL) is one of the circulating proteins with an average size of 7–12 nm and a density of 1.063–1.21 g/mL. In addition to reversing cholesterol transport (RCT), HDL possesses additional functions such as anti-oxidant, anti-inflammatory, anti-thrombotic, and immune-regulating activities [3]. HDL cholesterol (HDL-C) is generally deemed “good cholesterol” and Kjeldsen et al. have reported an inverse association between HDL-C and cardiovascular risk (CVR) [4]. However, epidemiological studies have revealed that, in individuals with CVD, the relationship between HDL-C and mortality risk was a U-shaped curve, with both extremely high and low concentrations being linked to an increased death risk [5,6]. Additionally, Mendelian randomization studies also suggested that the causal role between HDL-C and CVD was unclear [7,8]. What is more, niacin [9] and cholesteryl ester transfer protein (CETP) inhibitors [10] increased HDL-C but did not show clinical benefit. Thus, the “HDL-C hypothesis” has been turned into an “HDL function hypothesis” [11].

HDL’s cardioprotective effect largely depends on its cholesterol efflux capacity (CEC). Inverse correlations between CEC and CVD have been found in cross-sectional studies [12,13] and prospective studies [14,15], independently of the HDL-C level. However, in patients with diabetes on hemodialysis, Kopecky suggested that CEC did not predict CVR [16]. One possible reason is the heterogeneity of HDL. HDL consists of many proteins and lipids and can be further classified into different subclasses based on density, size, charge, and shape. Therefore, the HDL’s cardioprotective role may vary among subclasses.

In this review, we aim to summarize HDL subclasses and their clinical predictive values, and the effects of lipid lowering drugs and nonpharmacological approaches (exercise training, dietary omega fatty acids, and low-density lipoprotein apheresis) on HDL subclasses. Finally, the different characteristics of HDL subclasses as drug vehicles will be elaborated on. We hope this review could offer a useful reference for predicting CVR and assessing therapeutic approaches.

## 2. Different Nomenclatures of HDL Subclasses

HDL subclasses are classified by different separation methods via measuring different physical and chemical properties of HDL (Table 1) [17]. Ultracentrifugation (UC) is the gold standard technique. Based on density, HDL can be split into two subclasses: lipid-enriched HDL2 particles with a low density of 1.063–1.125 g/mL and protein-enriched HDL3 particles with a high density of 1.125–1.21 g/mL [18]. Then, non-denaturing polyacrylamide gradient gel electrophoresis (ND-PAGGE) further separates HDL2 and HDL3 into five subclasses based on size: HDL3c (7.2–7.8 nm), HDL3b (7.8–8.2 nm), HDL3a (8.2–8.8 nm), HDL2a (8.8–9.7 nm), and HDL2b (9.7–12.9 nm) [19]. Using nuclear magnetic resonance (NMR), small HDL (7.3–8.2 nm), medium HDL (8.2–8.8 nm), and large HDL (8.8–13.0 nm) are listed by size from small to large [20]. An ÄKTA fast protein liquid chromatography (FPLC) system also divides HDL into three subclasses according to size: small HDL (fractions 33–36), medium HDL (fractions 29–32), and large HDL (fractions 25–28) [21]. Depending on whether or not apolipoprotein (apo) A-II is present, HDL may be separated into two subclasses by immunodiffusion: LpA-I (containing only apoA-I) and LpA-I:A-II (containing both apoA-I and apoA-II) [22]. Agarose gel electrophoresis separates HDL into two subclasses based on HDL’s surface charge and shape: α-migrating particles and pre-β-migrating particles [20]. Furthermore, according to different charge and size, two-dimensional non-denaturing polyacrylamide gradient gel electrophoresis (2D-PAGGE) can be used to extract ten HDL subclasses: per-β1 HDL (5.6 nm), pre-α4/α4 HDL (7.4 nm), pre-α3/α3 HDL (8.1 nm), pre-α2/α2 HDL (9.2 nm), pre-α1/α1 HDL (11.0 nm), per-β2 HDL (>11.0 nm) [23]. As shown in Figure 1, pre-α HDL has a similar size to α HDL. However, pre-α HDL has a faster migration rate but lower amounts than α HDL and does not contain apoA-II [24]. Among these ten HDL subclasses, only α2 and α3 contain apoA-II [25]. Also, based on charge and size, Niisuke et al. utilized a 3D separation to identify five HDL subclasses: pre-β1 HDL (5.6 nm), α4 HDL (7.7 nm), α3 HDL (8.4 nm), α2 HDL (9.4 nm), and α1 HDL (11.0 nm) [26].

HDL subclasses have many nomenclatures defined by different methodologies, each with its own advantages and disadvantages. As a gold standard for lipoprotein separation, UC allows for isolating HDL and other lipoproteins in the meantime. However, UC requires a long procedure, and high salt concentrations and huge shear forces can strip proteins from HDL [27]. Compared to UC, FPLC is a relatively gentle method that does not affect proteins on HDL, but it can result in HDL being easily contaminated by co-eluted plasma proteins such as albumin [28]. ND-PAGGE is a sensitive approach for quantifying the size distribution of HDL subclasses. However, this method cannot separate pre-β HDL and requires specialized experimental instruments not commonly found in routine clinical laboratories [29]. The 2D-PAGGE is another sensitive way that combines ND-PAGGE with agarose gel electrophoresis, allowing for the successful separation of pre-β HDL [30]. However, this technique is limited to specialized laboratories and can overestimate pre-β concentrations in plasma [31]. An NMR analysis is precise and efficient for quantifying HDL subclasses, but it cannot detect pre-β HDL [32].

To make up for limitations, scientists continue to improve upon these methods. For example, asymmetric-flow field-flow fractionation (AF4) has been deemed a suitable alternative to FPLC, allowing for separating five HDL subclasses in the 7–18 nm range [33]. Zheng et al. even used a combined approach of UC and FPLC to separate HDL into four subclasses at different sizes [34]. According to the specific research purpose, researchers should reasonably choose an appropriate method to obtain HDL subclasses. For example, if the aim is to study the association of HDL subclasses with CVR in a population, UC and NMR are recommended because they are convenient for clinical usage. If the aim is to study the proteome and function of HDL, the combination of UC and FPLC is recommended because this combined approach can minimize other non-specific proteins’ interference. Overall, each separation has inherent characteristics. To better understand HDL subclasses, the HDL subclasses’ separations still need to be improved in the future.

## 3. Cardiovascular Risk (CVR) Prediction by HDL Subclasses

HDL2/HDL3 are the most widely studied subclasses in clinical research. As previously mentioned, total HDL can be further divided into HDL2 and HDL3 based on density. This allows researchers to independently assess the predictive value of both HDL2 and HDL3 in relation to CVD to determine which subclass has a stronger association. However, the question remains: which subclass is primarily responsible for the inverse association between HDL and CVR? Is it HDL2 or HDL3? There are three main viewpoints: (1) HDL2 is the strongest predictor of CVR [35,36,37]; (2) HDL3 is the main determinant of the negative association between HDL and CVR [38,39,40]; (3) compared to total HDL-C, neither HDL2 nor HDL3 provides additional predictive value for CVR [41,42]. Below, we summarize clinical studies supporting each of these viewpoints.

First, there are studies that support the first opinion. In 1966, Gofman et al. firstly showed that both HDL2 and HDL3 separated by UC were inversely associated with CVD, and the association was stronger for HDL2 than HDL3 [35]. Later, Salonen et al. conducted a prospective study in eastern Finnish men (*n* = 1799) to investigate the association between HDL2 and HDL3 with the risk of acute myocardial infarction (AMI) [36]. The study confirmed a strong inverse association between HDL2 and the risk of AMI, while the value of HDL3 remains unclear. Similarly, in a cross-sectional study with 115 men undergoing coronary angiography, Drexel and colleagues explored the association between HDL subclasses and the severity of CVD through a plasma lipid profile analysis [37]. The results showed that HDL2 was the strongest independent predictor of CVD, while smaller differences were found for HDL3 and total HDL-C.

Next, other scientists hold the second opinion. In two prospective cohorts, the TRIUMPH study (*n* = 2465) and the IHCS study (*n* = 2414), Martin et al. measured patients’ HDL subclasses by UC. Their findings indicated that HDL3, rather than HDL2 or total HDL-C, was inversely associated with mortality and myocardial infarction (MI) risk [38]. In Jackson Heart (*n* = 4114) and Framingham Offspring (*n* = 818) Cohort Study, Joshi et al. also concluded that HDL3, rather than HDL2, was primarily responsible for the inverse association between HDL-C and CVD [39]. Albers and co-workers conducted a secondary analysis of the AIM-HIGH clinical trial and found that HDL3 may drive the inverse association between HDL-C and CVD, while HDL2 did not predict the occurrence of cardiovascular events [40].

At last, there is the third opinion. In the Caerphilly and Speedwell prospective studies, Lamarche et al. found that although HDL3 is more strongly associated with CVD than HDL2, HDL3 did not show better predictive value than total HDL-C [41]. In 2012, Superko et al. conducted a systematic review of 80 published studies [42]. They concluded that, in identifying individuals with high CVR, HDL2 and HDL3 did not have more clinical benefit compared to HDL-C. They suggested that, in addition to HDL2 and HDL3, other HDL subclasses separated by methods like NMR or ND-PAGGE were also worth investigating.

Consequently, in addition to HDL2/HDL3, subsequent studies have explored the small/medium/large HDL subclasses as well. For example, McGarrah et al. separated small/medium/large HDL subclasses by NMR and investigated their association with all-cause mortality in a high-risk cardiovascular population (*n* = 3972) [43]. In this study, HDL particles had a stronger inverse association with mortality compared to HDL-C, with this relation being totally attributed to small and medium HDL, while large HDL showed no significant association. Similarly, Silbernagel et al. followed up with 2290 participants referred for a coronary angiograph and analyzed their small/medium/large HDL by NMR [44]. The findings indicated that small HDL was the primary factor driving the inverse association between HDL and cardiovascular mortality. In patients with heart failure, Potocnjak and Hunter et al. reported that small HDL was a strong and independent predictor of mortality [45,46]. Using NMR, Harbaum et al. categorized HDL into two main subclasses (small and large) and found that small HDL was the strongest prognostic factor in patients with pulmonary arterial hypertension [47]. Similarly, Duparc and colleagues also had two main HDL subclasses (small and large) and concluded that small HDL has excellent predictive value in patients with CVD [48]. To investigate the relationship between HDL subclasses and coronary plaque stability, Wang et al. divided ten HDL subclasses into three categories (small, medium, and large) [49]. They concluded that the small HDL subclass might represent the cardioprotective activity of HDL, as it was significantly positively associated with fibrous cap thickness. Varela et al. enrolled patients with ischemic stroke and reported that the ratio of large/small HDL was positively correlated with the unfavorable outcomes [50].

In Table 2, we list more details of the above clinical studies. Although most studies indicate that the small HDL subclass primarily plays the cardioprotective role, other controversial points are still worth considering. One possible explanation for the inconsistent results is that HDL not only changes shape and size as it is metabolized in the body, but the proteins and lipids it carries also transfer between different particles. It is challenging to identify a specific HDL subclass that consistently predicts outcomes across populations with different underlying diseases. Moreover, various methods for isolating HDL subclasses are used in different clinical studies, and further research is still needed to explain the conflicting results.

## 4. Effects of Hypolipidemic Drugs on HDL Subclasses

To reduce CVR, multiple hypolipidemic drugs focusing on low-density lipoprotein cholesterol (LDL-C) lowering have been developed, including statins, fibrates, CETP inhibitors, niacin, and proprotein convertase subtilisin/kexin type 9 inhibitors (PCSK9-I). Initially, the purpose of hypolipidemic drugs was to reduce LDL-C, with statins being the most common type, followed by the PCSK9-I. Nonetheless, fully adhering to statin therapy and achieving current LDL-C targets only result in an ~30% decrease in the risk of major cardiovascular events [51]. Hence, investigators turned to other types of lipoproteins to further decrease CVR, with HDL being a particularly attractive target. The concept that drugs designed to boost HDL-C can reduce cardiovascular events was supported by epidemiologic studies [52]. Although the main antiatherosclerosis targets of these hypolipidemic drugs differ, studies have found that they all affect the distribution of HDL subclasses.

### 4.1. LDL-C Lowering Drugs—Statins and PCSK9-I

#### 4.1.1. Statins Increase HDL Size

Statins, which are 3-hydroxy-3-methyl-glutaryl (HMG)-CoA reductase inhibitors, act primarily by increasing the expression of low-density lipoprotein (LDL) receptors, resulting in the elimination of LDL. Additionally, statins modestly increase HDL by 5% to 10% [53]. As shown in Table 3, Tomas and Franceschini et al. isolated HDL into LpA-I and LpA-I/A-II subclasses and found no significant difference in percentage distributions or concentrations between the statin and placebo groups [54,55]. Using ND-PAGGE, Asztalos et al. identified eight HDL subclasses [56,57]. They found that patients who received atorvastatin treatment showed higher pre-α1 and α1 and lower α3 HDL [56]. In New Zealand white rabbits, atorvastatin administration resulted in a similar trend, with a shift in HDL particle size towards larger ones [58]. The increase in HDL size may be due to reduced CETP activity [59], with atorvastatin mainly reducing the cholesteryl ester (CE) acceptor level [59], while simvastatin mainly reduces the plasma CETP mass [60].

#### 4.1.2. PCSK9-I Increases HDL Size

PCSK9-I is a novel hypolipidemic drug, which decreases LDL-C levels by increasing the amount of LDL receptors, thereby promoting LDL-C metabolism [61]. Arsenault et al. found that PCSK9-I not only significantly reduces LDL-C levels but also regulates the physicochemical characteristics of HDL [62]. Limited studies have investigated the PCSK9-I’s effect on HDL. Kalogirou et al. found that small HDL was specifically decreased by ezetimibe monotherapy, whereas other HDL subclasses remained unchanged (Table 3) [63]. Other studies showed that in addition to the decrease in small HDL, treatments with bococizumab [64] and alirocumab [65] significantly increased large HDL. In individuals with hypercholesterolemia who received PCSK9-I, Ingueneau et al. found that HDL size tended to marginally rise [66]. Thus, the size of HDL subclasses can increase after receiving PCSK9-I.

**Table 3 ijms-25-07856-t003:** Effects of drugs that decrease low-density lipoprotein cholesterol (LDL-C) levels on high-density lipoprotein (HDL) subclasses.

Drug	HDL Subclasses	Changes after Treatment	Reference
Simvastatin	LpA-I, LpA-I/A-II	No significant change	Tomas, M. et al. 2000 [54]
Atorvastatin	pre-β1, pre-β2, α1, α2, α3, pre-α1, pre-α2, pre-α3	α1↑ α3↓	Asztalos, B. F. et al. 2002 [57]
Atorvastatin, Simvastatin, Pravastatin, LovastatinFluvastatin	pre-β1, pre-β2, α1, α2, α3, pre-α1, pre-α2, pre-α3	pre-α1↑ (each statin) α1↑ (except fluvastatin) α3↓ (only atorvastatin and simvastatin)	Asztalos, B. F. et al. 2002 [56]
Simvastatin	Subclass 1: LpA-I, LpA-I:A-II Subclass 2: Small HDL, Medium HDL, Large HDL	No significant change	Franceschini, G. et al. 2007 [55]
Ezetimibe	Small HDL, Medium HDL, Large HDL	Small HDL↓	Kalogirou, M. et al. 2007 [63]
Alirocumab	Small HDL, Medium HDL, Large HDL	Large HDL (44.6%) ↑ Medium HDL (17.7%) ↑ Small HDL (2.8%) ↑	Koren, M. J. et al. 2015 [65]
Bococizumab	Small HDL, Medium HDL, Large HDL	Large HDL↑ Small HDL↓	Wan, H. et al. 2017 [64]
Ezetimibe	Small HDL, Medium HDL, Large HDL, Extra Large HDL	HDL size slightly increases	Ingueneau, C. et al. 2020 [66]

↑: moderately significant increase; ↓: moderately significant decrease.

### 4.2. HDL-C Elevating Drugs—Fibrates, Niacin, and CETP Inhibitors

Since the residual risk still exists after LDL-C reaches guideline-recommended levels, the strategy to raise HDL-C has begun to attract attention. So far, three kinds of medications have been developed to raise HDL-C levels: fibrates, niacin, and CEPT inhibitors. Fibrates and niacin can increase HDL-C levels by up to 20% and 30%, respectively [53]. Additionally, CETP inhibitors are also promising agents for raising HDL-C levels, but the increase degree varies from drug to drug [67]. As the first CETP inhibitor, torcetrapib increased HDL-C by up to 70% [68]. Dalcetrapib, a relatively mild CETP inhibitor, was reported by Kausik et al. to raise HDL-C by up to 30% [69]. Lincoff et al. found that evacetrapib increased HDL by approximately 133.2% [10]. In the DEFINE trial, Christopher reported that anacetrapib increased HDL by 138.1% [70]. The latest CETP inhibitor, obicetrapib, resulted in a 96–140% increase in HDL-C [71].

#### 4.2.1. Fibrates Decrease HDL Size

The primary role of fibrates in clinical practice is the management of hypertriglyceridemia, but these agents can also raise HDL-C to some extent [72]. A study conducted in patients with hypertriglyceridemia showed that the post-treatment level of pre-β1 HDL was twice as high as in the control group, and only the HDL2b showed a substantial decrease following bezafibrate administration [73]. This phenomenon may result from bezafibrate’s heightened lipase activity, causing HDL2, which is high in triglycerides (TGs), to produce pre-β1 HDL through the influence of hepatic lipase [74]. This results in the conversion of large HDL into smaller ones. In addition, fenofibrate has been shown to significantly increase HDL3 levels and decrease HDL2 (2a, 2b) levels in hyperlipidemic patients [75]. In dyslipidemic patients, Franceschini et al. found that fenofibrate could lead to a shift in HDL from large to small [55]. Similarly, in large cohorts of two VA-HIT trials, different HDL subclasses (small, medium, and large [76]; and pre-β1/2, α1/2/3, and pre-α1/2/3 [77]) were chosen to confirm that gemfibrozil raised contents of the small-size HDL (α3, pre-α3) while decreasing the levels of large-size HDL (α1, α2). Unlike statins and PSCK9-I, the fibrates can decrease the size of HDL subclasses.

#### 4.2.2. Niacin Increases HDL Size

Niacin, similar to fibrates, increases HDL-C levels and reduces TG. As shown in Table 4, multiple studies have consistently found that niacin helped HDL mature into larger particles, leading to an increase in the average size of HDL [78,79,80,81,82]. Specifically, in patients with primary hypercholesterolemia, Morgan et al. found that extended-release niacin (ERN) increased large-size HDL subclasses (HDL2a, HDL2b) but did not affect small-size HDL subclasses (HDL3a, HDL3b, HDL 3c) [78]. Similarly, Franceschini and colleagues observed no change in the number of small HDL after ERN intervention, while the large HDL significantly increased [82]. Additionally, Lamon-Fava et al. also concluded that ERN promoted the maturation of HDL into larger particles [80]. Being partially different from Morgan and Franceschini’s findings, Kuvin [79] and Jafri [81] et al. found that ERN not only significantly increased the large HDL but also significantly reduced the small HDL. Although the specific mechanism is unclear, a decrease in CETP activity due to the decline in TG concentration may be a contributing factor [80].

#### 4.2.3. CETP Inhibitors Increase HDL Size

As CETP transfers CE from HDL to apoB-containing particles, it reduces the cholesterol levels in HDL [88]. Therefore, CETP inhibition is considered an effective approach to increase HDL-C levels. Information about the impact of CETP inhibitors on HDL subclasses is limited (Table 4). Ballantyne et al. used two separation methods (NMR and ND-PAGGE) to analyze dalcetrapib’s effect on HDL subclasses: NMR showed a higher proportion of large HDL, and ND-PAGGE showed HDL2a and HDL2b enrichment [83]. Krauss divided HDL into five subclasses (HDL3c, HDL3b, HDL3a, HDL2a, HDL2b) and found that anacetrapib significantly increased large HDL2b [84]. Chen et al. found that evacetrapib decreased small HDL and pre-β1 HDL, and increased large HDL and medium HDL [87]. Obicetrapib (TA-8995), a novel CETP inhibitor, is expected to be the first agent in clinical use [71]. In a study using 2D-PAGGE, an increase in very small pre-β1 (36%) was found in patients with dyslipidemia after TA-8995 treatment, but very large α1 (350%) and pre-β2 (66%) HDL particles increased more [86]. A similar result was observed in the study of Nicholls et al. [85].

Although CETP inhibitors have been expected as a potential new way in preventing CVD, large clinical trials have showed disappointing results. Because of off-target effects, torcetrapib activated the renin angiotensin aldosterone system, leading to side effects such as high aldosterone and cortisol levels and electrolyte abnormalities [89]. Dalcetrapib and evacetrapib did not exhibit torcetrapib-like toxicity but were terminated early due to clinical futility [90,91]. Anacetrapib was associated with a significant 9% reduction in cardiovascular events. However, after stopping administration, prolonged lipid effects were observed, leading to adipose tissue accumulation in subjects. Thus, anacetrapib was finally not approved by regulators [92]. New CETP inhibitors are still being evaluated in large clinical trials. If there is still little cardiovascular benefit after a significant increase in HDL-C levels, it is crucial to carefully consider how to increase HDL subclasses that are truly cardioprotective.

## 5. Effects of Nonpharmacological Approaches on HDL Subclasses

Exercise training, omega fatty acids, and LDL apheresis have been investigated as nonpharmacological strategies to prevent or treat CVD. As the focus of research has shifted from the quantity to the quality of HDL, more studies are exploring whether these nonpharmacological approaches can alter HDL subclasses.

### 5.1. Exercise Training Increases HDL Size

Exercise training is a well-established way to reduce CVR, partly due to its effects on HDL. Greene and co-workers studied the impact of exercise training on HDL subclasses in both obese men and women [93]. In men, they found that increases in HDL-C were mediated by a rise in HDL2b. While HDL-C levels did not change in women, there was a shift in HDL subclasses from HDL3 to HDL2a and HDL2b after exercise training. In a meta-analysis, Sarzynski et al. analyzed 1555 adults from six studies and found that exercise training increased the large HDL levels while decreasing medium HDL levels [94]. Later, Rodriguez-Garcia et al. enrolled metabolically healthy obese women in a hypocaloric diet and exercise program. Their results showed that intensive lifestyle modification did not increase large HDL, but the overall atherogenic situation still improved due to significant reductions in small and medium HDL [95]. By comparing changes in small/medium/large HDL before and after exercising, Woudberg and colleagues found that the distribution of large HDL did not change, while the distribution of small HDL significantly decreased in the exercise group [96]. In a recent study, the effect of exercise training on HDL subclasses was investigated especially in older women [97]. The authors concluded that although total HDL concentrations decreased, the size of HDL subclasses increased in the training group. Collectively, regardless of whether total HDL concentration increased, the size of HDL subclasses consistently tended to be larger after exercise training. Large-size HDL subclasses have been considered healthier than smaller ones because they have a stronger cholesterol-carrying capacity [98].

### 5.2. Omega Fatty Acids Increase HDL Size

Omega fatty acids can inhibit the synthesis of lipids and lipoproteins in the liver, thereby reducing plasma cholesterol and TG, which is beneficial for maintaining cardiovascular health. Similar to exercise training, multiple studies have found that increasing dietary omega fatty acids is significantly associated with favorable changes in HDL subclasses. For instance, Wooten et al. investigated the effect of short-term omega fatty acid supplementation on HDL subclasses in sedentary men. They found that omega fatty acids caused a shift in HDL subclasses from smaller HDL3a and HDL3b to larger HDL2b and HDL2a [99]. In healthy young adult twins, Bogl et al. also found that omega fatty acids increased HDL2b and decreased HDL3a and HDL3b [100]. Grytten et al. separated small/medium/large HDL using NMR and observed that large HDL increased and small HDL decreased after omega fatty acid supplementation [101]. Moosavi reported that, compared to the placebo group, the content of large HDL significantly increased and small HDL significantly decreased in the omega fatty acid group [102]. The reason for this phenomenon is a decrease in CETP activity, which reduces the exchange of cholesterol from HDL to very-low-density lipoprotein (VLDL) and LDL, consequently increasing HDL subclasses with a large size [103]. The increase in large HDL subclasses suggests potential cardioprotective effects of omega fatty acids, consistent with the effects of exercise training. Therefore, in addition to hypolipidemic drugs, a combination of exercise and dietary omega fatty acids should be considered for the more effective prevention and treatment of CVD.

### 5.3. LDL Apheresis Reduces All Sizes of HDL Subclasses

When conventional cholesterol-lowering therapies are ineffective, LDL apheresis is considered a treatment option of a last resort, especially for patients with familial hypercholesterolemia. Although LDL apheresis can significantly decrease apoB-containing particles by 60%, it has been reported to reduce HDL-C by 10% to 15% [104]. Orsoni et al. categorized HDL into five subclasses (HDL2b, HDL2a, HDL3a, HDL3b, HDL3c) and found that small HDL3b and HDL3c were preferentially removed from all HDL subclasses by LDL apheresis [105]. Lappegard et al. explored changes in HDL subclasses when switching treatment from LDL apheresis to PCSK9-I. During LDL apheresis, small, medium, and large HDL all reduced non-significantly [106]. However, the decreased concentrations of small HDL were the most (from 12.0 ± 4.6 mg/dL to 5.0 ± 1.7 mg/dL) among all HDL subclasses. Thus, LDL apheresis can simultaneously reduce all sizes of HDL subclasses, primarily affecting small-size HDL subclasses. However, the research is quite limited, and more studies are needed to further explore the effects of LDL apheresis on HDL subclasses.

## 6. Differences between d-rHDL and s-rHDL in Clinical Applications

HDL is a natural nanoparticle capable of transporting cholesterol in mammalian circulation. Due to their nanoscale size, long half-life, and abundance of lipids and proteins, HDL has a strong potential as a delivery vehicle. For example, Lin et al. used natural HDL isolated from human plasma to encapsulate 10-hydroxycamptothecin (HCPT) [107]. To overcome the blood–brain barrier and blood–brain tumor barrier, they also added dual modifications with T7 and ^d^A7R peptide ligands to the HDL. Ultimately, this natural HCPT-loaded T7/^d^A7R-HDL was able to target glioma cells and demonstrated excellent anti-glioma efficacy. Additionally, David et al. replaced the hydrophobic core of HDL with inorganic nanocrystals, creating novel HDL-based contrast agents successfully used in computed tomography, magnetic resonance, and fluorescence imaging [108]. However, purifying HDL from human plasma carries the risk of infection and is quite cumbersome. To prepare enough high-quality HDL for in vivo therapeutic use, researchers have used industrial methods to synthesize recombinant high-density lipoprotein (rHDL), such as the sodium cholate dialysis method, sonication method, and thermal cycling method [109]. The rHDL offers advantages of high stability, long storage time, ease of synthesis, and consistent quality from batch to batch. Studies mainly divide rHDL into two subclasses, discoidal rHDL (d-rHDL) and spherical rHDL (s-rHDL), each with its own characteristics.

The d-rHDL has advantages: it requires fewer raw materials for preparation, potentially reducing synthesis expense and simplifying the procedure [110]. Additionally, the bilayer arrangement of d-rHDL enables the incorporation of membrane proteins such as SR-BI, while spherical particles are not well suited for accommodating transmembrane proteins [111].

The s-rHDL also offers benefits: its specific hydrophobic core enables the encapsulation of medications both on the membrane and within the particle’s core, creating a water-resistant environment for the payloads [112]. Furthermore, spherical HDL is more stable than discoidal HDL, especially when plasma remodeling enzymes exist. For example, Zhang et al. created d-rHDL containing tanshinone IIA (TA) and found that TA leaked from TA-d-rHDL during the transition from a discoidal to spherical shape [113]. In contrast, the s-rHDL containing paclitaxel (rHDL/PTX) created by Walter et al. maintained superior stability even after 6 months of storage [114]. Additionally, the synthetic methods for s-rHDL are more diverse than those for d-rHDL [110]. Thus, the s-rHDL has been widely used in some clinical treatment scenarios, such as PTX [114], valrubicin [115], epidermal growth factor [116], and nucleic acids [117].

Currently, rHDLs have been used in clinical trials, such as ETC-216, MDCO-216, CER-001, and CSL-111, but the outcomes have been unsatisfactory due to a poor curative effect or side effects. The infusion of pre-β HDL mimetic CER-001 had no effect in statin-treated people with ACS [118]. However, CER-001 has shown therapeutic potential in reducing inflammation and organ damage in patients with sepsis [119]. In the latest multicenter, randomized, double-blind, placebo-controlled AEGIS-II trial, Thomas et al. found that patients receiving CSL112 (a reconstituted human apoA-I) infusion had a lower rate of cardiovascular mortality, even though the primary composite endpoint was insignificantly reduced [120]. However, the study is an exploratory analysis, and further prospective studies are needed to validate these observations.

## 7. Conclusions

As the credibility of HDL-C diminished, it is crucial to find a new biomarker that can replace HDL-C for further investigation [121]. Nonetheless, current evidence regarding the association of specific HDL subclasses with CVR is conflicting and confusing. Because of the different methods for separating HDL subclasses and the dynamic HDL metabolism, multiple static laboratory measurements have inherent limitations [122]. For a specific purpose, scientists conduct experiments based on different subjects, isolation techniques, and analytical methods, making it unrealistic to compare HDL subclasses separated by different methods. For instance, comparing subclasses created by density-based approaches to those created by size-based approaches is not feasible. Not only that, HDL2 and HDL3 levels are primarily affected by the combined activity of four enzymes (LCAT, CETP, hepatic triglyceride lipase, and lipoprotein lipase) in the plasma. Whether these enzymes are genetically influenced differently in the studied populations is unknown. Other conditions, such as diabetes and hypertension, might also significantly influence it [123]. This potential discrepancy emphasizes the need to study the association between genetic variations and subclasses’ distribution in different populations [124].

Therefore, it remains uncertain whether the cardioprotective benefits of HDL are due to one or both of its subclasses, such as HDL2/HDL3 or small/medium/large HDL. Although the proatherogenic and antiatherogenic properties of HDL subclasses are still critical, the changes and complexity observed in HDL subclasses may play a valuable role. What is more, due to specific physicochemical properties, HDL particles can cluster into different subclasses instead of being randomly distributed throughout the plasma. This process in turn causes dynamic changes in lipoproteins among HDL subclasses [125]. For example, an increase in apoA-I in HDL that contains apoC3 was proven to be the reason why CETP inhibitors failed to lower the risk of CVD [126]. CETP inhibitors elevate lipid contents and HDL particle size, which provides a more suitable living environment for apoA-I. The understanding of dynamic changes in lipoproteins among HDL subclasses will help scientists to interrelate the complex methodologies into an overall functional construct. The variations of HDL subclasses indicate our poor understanding of HDL biology.

In conclusion, since both predictive biomarkers and drug delivery agents require highly precise targets, larger-scale studies are warranted to figure out the changes in HDL subclass distribution. An advanced HDL subclass analysis may identify better predictors of CVR. Once these targets are identified, profound insights could aid in developing specific drug therapies to reduce CVR and designing more effective drug delivery systems for rHDL.

## Figures and Tables

**Figure 1 ijms-25-07856-f001:**
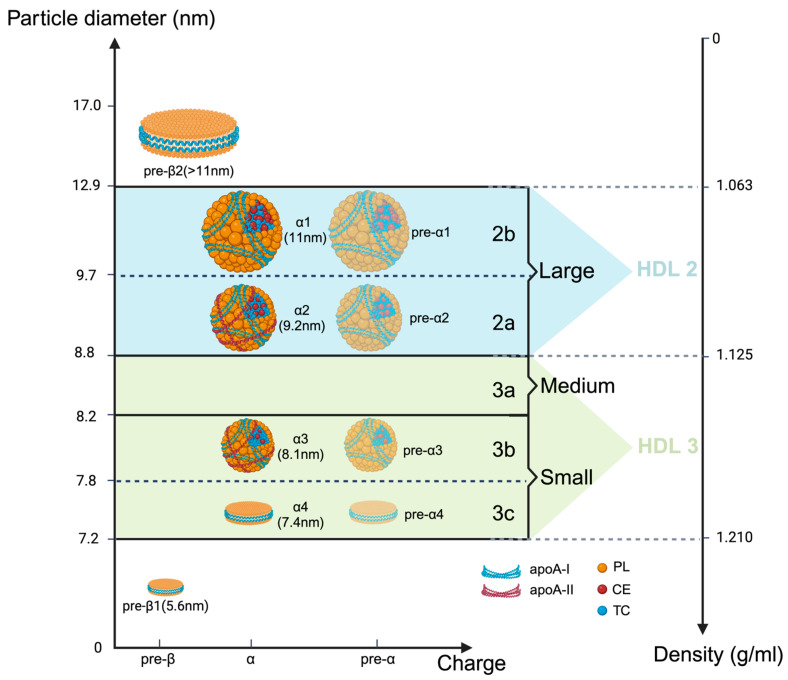
The comparison of different nomenclatures for HDL subclasses. Summarizing all the methods for naming HDL subclasses based on charge, size, density, and shape, as well as the presence or absence of apolipoprotein A-II. Based on charge (the abscissa axis), three major HDL subclasses have been identified: pre-β, α, and pre-α. Then, these three subclasses can be further divided into ten subclasses because of different diameter (the left vertical axis): very small discoidal pre-β1 HDL (5.6 nm), small discoidal α4 and pre-α4 HDL (7.4 nm), small but spherical α3 and pre-α3 HDL (8.1 nm), medium spherical α2 and pre-α2 HDL (9.2 nm), large spherical α1 and pre-α1 HDL (11 nm), very large per-β2 HDL (>11 nm). Compared with pre-β1, pre-β2, and α4, which only contain apoA-I (the blue curves), phospholipid (the yellow globe), and free cholesterol, the α3, α2, and α1 contained an additional cholesterol ester (the red globe) and triglyceride (the blue globe) core. Pre-α HDL is adjacent to the α counterparts, corresponding to the lighter shaded part in this figure. Pre-α HDL had a faster migration rate but lower amounts than α HDL and did not contain apoA-II (the red curves). In other nomenclatures, the range of HDL3c and 3b correspond to small HDL, HDL3a corresponds to medium-sized HDL, and HDL2a and 2b correspond to large HDL. From small HDL to large HDL, the density gradually decreases (the right vertical axis). Larger HDL2 possesses a density of 1.063–1.125 g/mL and smaller HDL3 has a density of 1.125–1.21 g/mL.

**Table 1 ijms-25-07856-t001:** Classification of high-density lipoprotein (HDL) subclasses by different physical and chemical properties.

Isolation Techniques	HDL Subclasses	Density
UC	HDL2	1.063–1.125 g/mL
	HDL3	1.125–1.210 g/mL
		Size
ND-PAGGE	HDL3c	7.2–7.8 nm
	HDL3b	7.8–8.2 nm
	HDL3a	8.2–8.8 nm
	HDL2a	8.8–9.7 nm
	HDL2b	9.7–12.9 nm
NMR	Small HDL	7.3–8.2 nm
	Medium HDL	8.2–8.8 nm
	Large HDL	8.8–13.0 nm
		Charge and Shape
Agarose gel electrophoresis	per-β HDL	spherical
	α HDL	discoidal
		Charge and Size
2D-PAGGE	per-β1 HDL	5.6 nm
	pre-α4/α4 HDL	7.4 nm
	pre-α3/α3 HDL	8.1 nm
	pre-α2/α2 HDL	9.2 nm
	pre-α1/α1 HDL	11.0 nm
	per-β2 HDL	>11.0 nm
3D separation method	pre-β1	5.6 nm
	α4	7.7 nm
	α3	8.4 nm
	α2	9.4 nm
	α1	11 nm
		Fraction Number
FPLC	Small HDL	33–36
	Medium HDL	29–32
	Large HDL	25–28
		Apolipoprotein Content
Electro-immunodiffusion	LpA-I	apoA-I
	LpA-I:A-II	apoA-I + apoA-II

apoA-I: apolipoprotein A-I; apoA-II: apolipoprotein A-II; FPLC: fast protein liquid chromatography; ND-PAGGE: non-denaturing polyacrylamide gradient gel electrophoresis; NMR: nuclear magnetic resonance; UC: ultracentrifugation; 2D-PAGGE: two-dimensional non-denaturing polyacrylamide gradient gel electrophoresis.

**Table 2 ijms-25-07856-t002:** Results of the main clinical studies examining the association between high-density lipoprotein (HDL) subclasses and cardiovascular disease (CVD).

Study Population	Methodology	Main Findings	RR/HR/r and *p* Value	Reference
1799 patients with IHD	UC	HDL2 ^1^ levels have inverse associations with the risk of AMI, whereas the role of HDL3 ^2^ remains equivocal.	^1^ *p* < 0.001 ^2^ *p* = 0.07	Salonen, J. T. et al. 1991 [36]
1115 men undergoing coronary angiography	UC	HDL2 ^1^ to be the strongest predictor of extent of CVD, while HDL3 ^2^ holds weaker significance.	^1^ *p* = 0.0009 ^2^ *p* = 0.0069	Drexel, H. et al. 1992 [37]
83 patients with IHD 861 men without IHD	UC	HDL2 and HDL3 contribute equally to IHD risk, but the contribution of HDL2 ^1^ was statistically significant, whereas it did not reach significance for HDL3 ^2^.	^1^ RR = 0.84 (0.74–0.95) ^2^ RR = 0.87 (0.69–1.11)	Lamarche, B. et al. 1997 [41]
TRIUMPH: 2465 patients with AMI IHCS: 2414 patients undergoing coronary angiography	UC	Mortality or MI in patients undergoing angiography to be strongly associated with lower HDL3, but not with HDL2. HDL3 was independently associated with higher risk in each cohort (TRIUMPH: with middle tertile as reference, Tertile 1 vs. 2, 2-year mortality ^1^; IHCS: with middle tertile as reference, Tertile 1 vs. 2, 5-year mortality ^2^).	^1^ HR = 1.57 (1.13–2.18) ^2^ HR = 1.55 (1.20–2.00)	Martin, S. S. et al. 2015 [38]
JHS: 4114 participants FOCS: 818 participants	UC	HDL3 ^1^, rather than HDL2 ^2^, drove the inverse association of HDL-C ^3^ with CHD.	^1^ HR = 0.76; *p* = 0.01 ^2^ HR = 0.88; *p* = 0.24 ^3^ HR = 0.79; *p* = 0.03	Joshi, P. H. et al. 2016 [39]
3094 participants who were already on statin therapy prior to enrollment in the trial	UC	HDL3 ^1^ was protective against CVD events, while HDL-C ^2^ and HDL2 ^3^ were not event-related.	^1^ HR = 0.84; *p* = 0.043 ^2^ HR = 0.96; *p* = 0.369 ^3^ HR = 1.07; *p* = 0.373	Albers, J. J. et al. 2016 [40]
3972 patients with suspicious IHD undergoing coronary catheterization	NMR	HDL-P ^1^ had a stronger inverse association with mortality than HDL-C ^2^. Large HDL ^3^ conferred greater risk and the sum of medium HDL ^4^ and small HDL ^5^ conferred less risk.	^1^ HR = 0.71; *p* < 0.0001 ^2^ HR = 0.93; *p* = 0.02 ^3^ HR = 1.03; *p* = 0.29 ^4^ HR = 0.73; *p* < 0.0001 ^5^ HR = 0.64; *p* < 0.0001	McGarrah, R. W. et al. 2016 [43]
2290 participants referred for coronary angiography	NMR	HDL-P ^1^ was inversely related to cardiovascular mortality, which was primarily mediated by small HDL ^2^, while the medium HDL ^3^ and large HDL ^4^ did not reach statistical significance.	^1^ HR = 0.55; *p* < 0.001 ^2^ HR = 0.60; *p* < 0.001 ^3^ HR = 0.85; *p* = 0.177 ^4^ HR = 0.75; *p* = 0.057	Silbernagel, G. et al. 2017 [44]
152 patients with AHF	NMR	A significant inverse association of small HDL ^1^ with mortality was observed, whereas large HDL ^2^ showed no significant association.	^1^ OR = 0.35; *p* < 0.001 ^2^ *p* = 0.353	Potocnjak, I. et al. 2017 [45]
Patients with HFrEF (*n* = 782) or HFpEF (*n* = 1004)	NMR	In both HFrEF ^1^ and HFpEF ^2^, small HDL was inversely associated with time to adverse events.	^1^ HR = 0.70; *p* < 0.0001 ^2^ HR = 0.73; *p* < 0.0001	Hunter, W. G. et al. 2019 [46]
204 patients with PAH	NMR	Only small HDL levels were independently prognostic in patients with PAH.	*p* < 0.05	Harbaum, L. et al. 2019 [47]
85 patients with CAD	ND-PAGGE	Small HDL ^1^ was positively associated with fibrous cap thickness, but medium HDL ^2^ and large HDL ^3^ did not show statistical correlation.	^1^ *r* = 0.243; *p* = 0.005 ^2^ *r* = −0.033; *p* = 0.706 ^3^ *r* = −0.127; *p* = 0.145	Wang, X. et al. 2019 [49]
214 participants with CAD	NMR	Small HDL ^1^ was inversely associated with cardiovascular mortality, while large HDL ^2^ did not reach statistical significance.	^1^ HR = 0.61; *p* = 0.001 ^2^ HR = 0.97; *p* = 0.82	Thibaut, D. et al. 2020 [48].
50 patients with ischemic stroke	ND-PAGGE	The large HDL ^1^ was significantly positively correlated with the unfavorable outcomes while the small HDL ^2^ showed negative correlation.	^1^ R = 0.3871; *p* = 0.0079 ^2^ R = −0.4217; *P* = 0.0035	Varela, L. M. et al. 2020 [50]

AHF: acute heart failure; AMI: acute myocardial infarction; CAD: coronary artery disease; CHD: coronary heart disease; CVD: cardiovascular disease; FOCS: Framingham Offspring Cohort Study; HDL: high-density lipoprotein; HDL-P: HDL particle; HFrEF: heart failure with reduced ejection fraction; HFpEF: heart failure with preserved ejection fraction; HR: hazard ratio; IHCS: Intermountain Heart Collaborative Study; IHD: ischemic heart disease; JHS: Jackson Heart Study; MI: myocardial infarction; ND-PAGGE: non-denaturing polyacrylamide gradient gel electrophoresis; NMR: nuclear magnetic resonance; PAH: pulmonary arterial hypertension; R: Spearman’s correlation coefficient; RR: relative risk; TRIUMPH: Translational Research Investigating Underlying disparities in acute Myocardial infarction Patient’s Health status; UC: ultracentrifugation.

**Table 4 ijms-25-07856-t004:** Effects of drugs that raise high-density lipoprotein cholesterol (HDL-C) levels on high-density lipoprotein (HDL) subclasses.

Drug	HDL Subclasses	Changes after Treatment	Reference
Bezafibrate	pre-β1, pre-β2, pre-β3, HDL2b, HDL2a, HDL3	pre-β1↑ HDL2b↓	Miida, T. et al. 2000 [74]
Fenofibrate	HDL2a, HDL2b, HDL3	HDL3↑ HDL2a, HDL2b↓	Sasaki, J. et al. 2002 [75]
Gemfibrozil	Small HDL, Medium HDL, Large HDL	Small HDL↑	Otvos, J. D. et al. 2006 [76]
Fenofibrate	Small HDL, Medium HDL, Large HDL	Small HDL↑ Large HDL↓	Franceschini, G. et al. 2007 [55]
Gemfibrozil	pre-β1, pre-β2, α1, α2, α3, pre-α1, pre-α2, pre-α3	α3, pre-α3↑ α1, α2↓	Asztalos, B. F. et al. 2008 [77]
ERN	HDL3c, HDL3b, HDL3a, HDL2a, HDL2b	HDL2a, HDL2b↑	Morgan, J. M. et al. 2003 [78]
ERN	Small HDL, Medium HDL, Large HDL	Large HDL↑ Small HDL↓	Kuvin, J. T. et al. 2006 [79]
ERN	pre-β1, pre-β2, α1, α2, α3, α4, pre-α1, pre-α2, pre-α3, pre-α4	α1, α2, pre-α1, pre-α2↑	Lamon-Fava, S. et al. 2008 [80]
ERN	Small HDL, Medium HDL, Large HDL	Large HDL↑ Small HDL↓	Jafri, H. et al. 2009 [81]
ERN	Small HDL, Medium HDL, Large HDL	Large HDL↑	Franceschini, G. et al. 2013 [82]
Dalcetrapib	Subclass 1: Small HDL, Medium HDL, Large HDL Subclass 2: HDL2a, HDL 2b, HDL 3a, HDL 3b, HDL 3c	Large HDL, HDL2a, HDL2b↑	Ballantyne, C. M. et al. 2012 [83]
Anacetrapib	HDL3c, HDL3b, HDL3a, HDL2a, HDL2b	HDL2b↑	Krauss, R. M. et al. 2012 [84]
Evacetrapib	pre-β1, pre-β2, α1, α2, α3, α4, pre-α1, pre-α2, pre-α3, pre-α4	α1 (208%) ↑↑ pre-α1 (174%) ↑↑ pre-β1 (45%) ↑	Nicholls, S. J. et al. 2015 [85]
TA-8995	pre-β1, pre-β2, α1, α2, α3, α4, pre-α1, pre-α2, pre-α3, pre-α4	α1 (350%) ↑↑↑ pre-β2 (66%) ↑↑ pre-β1 (36%) ↑ α3↓	Van Capelleveen, J. C. et al. 2016 [86]
Evacetrapib	pre-β1 HDL, Small HDL, Medium HDL, Large HDL	Large HDL, Medium HDL↑ Small HDL, pre-β1↓	Chen, Y. et al. 2019 [87]

ERN: extended-release niacin; ↑: moderately significant increase; ↓: moderately significant decrease; ↑↑: highly significant increase; ↑↑↑: extremely significant increase.

## Data Availability

No new data were created or analyzed in this study.

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
