# Peer review of "High-Density Lipoprotein Subclasses and Their Role in the Prevention and Treatment of Cardiovascular Disease: A Narrative Review"

_ijms, 2024, doi:10.3390/ijms25147856_

Round 1

Reviewer 1 Report

Comments and Suggestions for Authors

This is a review report about the importance of HDL subclasses in cardiovascular disease. The authors demonstrated there were several methods measuring HDL subclasses, and the different methods could lead to different roles of HDL subclasses in the development of cardiovascular disease. The report was addressed quite well. I have some minor comments.

# The authors presented some findings about the link between ischemic heart disease and HDL subclasses. How about the link of HDL subclasses with other etiologies such as heart failure, cerebrovascular disease or other vascular diseases?

# The authors demonstrated the pharamacologica effects on HDL subclasses. How about the effect of exercise training on the HDL subclasses? 

Comments on the Quality of English Language

No comment

Author Response

Comments 1: The authors presented some findings about the link between ischemic heart disease and HDL subclasses. How about the link of HDL subclasses with other etiologies such as heart failure, cerebrovascular disease or other vascular diseases?

Response 1: Thank you for your helpful suggestion. We searched for relevant studies on the association of HDL subclasses with other etiologies, adhering to the following criteria: (1) the studies must be clinical; (2) the study populations must have cardiovascular disease (excluding ischemic heart disease); and (3) classical isolation methods for HDL subclasses must be used. After screening, we added new representative clinical studies on the association of HDL subclasses with heart failure, pulmonary arterial hypertension, and ischemic stroke. As these studies utilized small/medium/large HDL subclasses, we have included them in paragraph 5 of part 3 (page 5-6, lines 187-191, 198-200). Additionally, the results have been rewritten in Table 2 (page 6-7, lines 210-220).

The revised text is shown below:

  • page 5, line 187-191

In patients with heart failure, Potocnjak and Hunter et al. reported that small HDL was a strong and independent predictor of mortality[45,46]. Using NMR, Harbaum et al. cat-egorized HDL into two main subclasses (small and large) and found that small HDL was the strongest prognostic factor in patients with pulmonary arterial hypertension[47]. ”

  • page 5-6, line 198-200

“Varela et al. enrolled patients with ischemic stroke and reported that the ratio of large/small HDL was positively correlated with the unfavorable outcomes[50]. 

Comments 2: The authors demonstrated the pharamacologica effects on HDL subclasses. How about the effect of exercise training on the HDL subclasses?

Response 2: Thank you for your helpful suggestion. Since exercise training is a non-pharmacological method, after describing the effect of drugs on HDL subclasses, we added the effect of non-pharmacological methods on HDL subclasses as a new part. The effects of exercise on HDL subclasses were carefully described in this section.

The revised text is shown below:

“5.1. Exercise training increases HDL size

Exercise training is a well-established way to reduce CVR, partly due to its effects on HDL. Greene and co-workers studied the impact of exercise training on HDL subclasses in both obese men and women. In men, they found that exercise-induced increases in HDL-C were mediated by a rise in HDL2b. While HDL-C levels did not change in women, there was a shift in HDL subclasses from HDL3 to HDL2a and HDL2b after training[95]. In a meta-analysis, Sarzynski et al. analyzed 1,555 adults from six studies and found that exercise training increased the large HDL levels while decreasing medium HDL levels [96]. Later, Rodriguez-Garcia et al. enrolled metabolically healthy obese women in a hypocaloric diet and exercise program. Their results showed that intensive lifestyle modification did not increase large HDL, but the overall atherogenic situation still im-proved due to significant reductions in small and medium HDL[97]. Similarly, Woudberg and colleagues randomly assigned 32 obese women to either an exercise group or a control group (no exercise). By comparing changes in small/medium/large HDL before and after exercising, they found that the distribution of large HDL did not change, while the distribution of small HDL significantly decreased in the exercise group[98]. In a recent study, the effect of exercise training on HDL subclasses was investigated especially in older women[99]. The authors concluded that although total HDL concentrations de-creased, the size of HDL subclasses increased in the training group. Collectively, re-gardless of whether total HDL concentration increased, the size of HDL subclasses con-sistently tended to be larger after excise training. Larger HDL subclasses have been considered healthier than smaller ones because they have a stronger cholesterol-carrying capacity[100].” (page 11, line 344-365)

Reviewer 2 Report

Comments and Suggestions for Authors

The authors submitted the narrative review in which they concentrated on the HDL subclasses and their role in CVD risk and cardioprotection. The aim of the study is clear, the manuscript has a logical structure, the tables and fgures are legible. The article is well referenced. The findings of the study seem to be impressive and have sufficient clinical significance. However, I would like to make some comments to discuss.

1. Although the authors described that CETP inhibitors could increase HDL size, they might extend this subsection and suggest whether the inconclusive impact on HDL particle is plausible cause of variable effect on CVD mortality.

2. Yet, the authors did not discuss an impact of apheresis on HDL.

Author Response

Comments 1: Although the authors described that CETP inhibitors could increase HDL size, they might extend this subsection and suggest whether the inconclusive impact on HDL particle is plausible cause of variable effect on CVD mortality.

Response 1: We think this suggestion is excellent. We have checked the literature carefully and added more references on CETP inhibitors into the 4.2.3 Part. The potential adverse effects of CETP inhibitors and controversies regarding this class of drugs have been discussed in the revised manuscript.

The revised text is shown below:

“Although CETP inhibitors have been expected as a potential new way to preventing CVD, large clinical trials have showed disappointing results. Because of off-target effects, torcetrapib activated the renin angiotensin aldosterone system, leading to side effects such as high aldosterone and cortisol levels and electrolyte abnormalities[91]. Dalcetrapib and evacetrapib did not exhibit torcetrapib-like toxicity but were terminated early due to clinical futility[92,93]. Anacetrapib was associated with a significant 9% reduction in cardiovascular events. However, after stopping administration, prolonged lipid effects were observed, leading to adipose tissue accumulation in subjects. Thus, anacetrapib was not approved by regulators finally[94]. New CETP inhibitors are still being evaluated in large clinical trials. If there is still little cardiovascular benefit after a significant increase of HDL-C levels, it is crucial to carefully consider how to increase HDL subclasses that are truly cardioprotective.” (page 10, line 327-338)

Comments 2:  Yet, the authors did not discuss an impact of apheresis on HDL

Response 2: Thank you for your helpful suggestion. Since apheresis is a non-pharmacological method, after describing the effect of drugs on HDL subclasses, we added the effect of non-pharmacological methods on HDL subclasses as a new part. The impact of apheresis on HDL subclasses were carefully described in this section.

The revised text is shown below:

“5.3. LDL apheresis reduces all sizes of HDL subclasses

When conventional cholesterol-lowering therapies are ineffective, LDL apheresis is considered a treatment option of last resort, especially for patient with familial hyper-cholesterolemia. Although LDL apheresis can significantly decrease apoB-containing particles by 60%, it has been reported to reduce HDL-C by 10% to 15%[106]. However, studies on the effects of LDL apheresis on HDL subclasses are limited. Orsoni et al. categorized HDL into five subclasses (HDL2b, HDL2a, HDL3a, HDL3b, HDL3c) and found that small HDL3b and HDL3c were preferentially removed from all HDL subclasses by LDL apheresis[107]. Lappegard et al. explored changes in HDL subclasses when switching treatment from LDL apheresis to PCSK9-I. During LDL apheresis, small, medium, and large HDL all reduced non-significantly[108]. However, the decreased concentrations of small HDL were the most (from 12.0 ± 4.6 mg/dL to 5.0 ± 1.7 mg/dL) among all HDL subclasses. Thus, LDL apheresis can simultaneously reduce all sizes of HDL subclasses, primarily affecting small-size HDL subclasses. However, the research is quite limited, and more studies are needed to further explore the effects of LDL apheresis on HDL subclasses.” (page 11-12, line 387-401)

Reviewer 3 Report

Comments and Suggestions for Authors

Greetings to the authors, my comments are as follows.

The title should be better thought out.

One other potential issue is that authors barely mention apolipoproteins in the manuscript although they are know to have an important role in CV disease, the authors should keep this aspect in mind and consult whatever literature they desire.

Table 1 under the range of physical and chemical properties the authors list a range of different characteristics under the same column, such as density and size, and units of measurement ranging from g/ml to nm under the same column, this makes the table rather hard to interpret and understand, the layout and columns and rows should be rethought in such a way that it is consistent and easy to interpret as this is the overall scope of a table.

lines 134 to 157 the authors talk between the studies and differences between HDL class 2 and 3. However they do it alternatively in a manor that should be better written. The way these paragraphs are presented right now is rather hard and unpleasant to read. The authors should mention the studies that have explored HDL2 and then proceed to mention the studies that explore HDL3 in an orderly manner, the way these paragraphs are right now switch between HDL2 and HDL3 again and again and is not very coherent.

Authors briefly mention ezetimibe which is a very used lipid lowering medication drug, since it is widely used, and authors have mentioned almost all the other classes of lipid lowering medication, perhaps ezetimibe deserves a paragraph of it's own.

Not only that but the authors do not mention other lipid modifying medication such as omega fatty acids or bile acid sequestrants.

The paragraph regarding niacin is rather short and lacking in detail.

The authors fail to mention the potential adverse effects of cetp inhibitors and controversies regarding this class of drugs.

Line 289 authors state "As a natural biological nanocarrier, HDL can transport specific contrast agents and drugs to target tissues, improving diagnosis and treatment efficiency" but they should revise this statement and be clearer with what practical applications does HDL have in such situations.

Comments on the Quality of English Language

English is good overall. Requires moderate editing. 

Author Response

Comments 1: The title should be better thought out.

Response 1: Thanks for your valuable advice. After revising the content of our manuscript in accordance with the reviewers' comments, we re-evaluated the entire article and drafted a new title: HDL Subclasses and Their Role in Prevention and Treatment of Cardiovascular Disease: A Review.”

Comments 2: One other potential issue is that authors barely mention apolipoproteins in the manuscript although they are know to have an important role in CV disease, the authors should keep this aspect in mind and consult whatever literature they desire.

Response 2: Thank you for your helpful suggestion. HDL subclasses are groups of particles that can be easily separated based on their physicochemical properties. After reviewing the literature, we found that the apolipoproteins, which perform specific functions in physiological processes, dynamically change among HDL subclasses. This discussion has been added to the revised version as follows.

“What’s more, due to specific physicochemical properties, HDL particles can cluster into different subclasses instead of randomly distributed throughout the plasma. This process in turn causes dynamic changes of lipoproteins among HDL subclasses[128]. For example, an increase of apoA-I in HDL that contain apoC3 was proved to be the reason why CETP inhibitors failed to lower the risk of CVD.[129] CETP inhibitors elevate lipid contents and HDL particle size, which provides a more suitable living environment for apoA-I. The understanding of dynamic changes of lipoproteins among HDL subclasses will help scientists to interrelate the complex methodologies into an overall functional construct. The variations of HDL subclasses indicate our poor understanding of HDL biology. ” (page 13, line 466-474) 

Comments 3: Table 1 under the range of physical and chemical properties the authors list a range of different characteristics under the same column, such as density and size, and units of measurement ranging from g/ml to nm under the same column, this makes the table rather hard to interpret and understand, the layout and columns and rows should be rethought in such a way that it is consistent and easy to interpret as this is the overall scope of a table.

Response 3: Thank you for your helpful suggestion. We have revised the presentation of Table 1. HDL subclasses separated by the same physical and chemical properties have been grouped together, and we have used subheadings to categorize them. The revised Table 1 can be found on page 4, line 133 to 138.

Comments 4: lines 134 to 157 the authors talk between the studies and differences between HDL class 2 and 3. However they do it alternatively in a manor that should be better written. The way these paragraphs are presented right now is rather hard and unpleasant to read. The authors should mention the studies that have explored HDL2 and then proceed to mention the studies that explore HDL3 in an orderly manner, the way these paragraphs are right now switch between HDL2 and HDL3 again and again and is not very coherent.

Response 4: Thank you for your helpful suggestion. We apologize for our unclear expression. In response, we have added more explanatory text at the beginning to enhance understanding (page 4-5, line 140-149), and we have structured the different studies into three paragraphs (page 5, line 150-177)

The revised text is shown below:

  • page 4-5, line 140-149

“HDL2/HDL3 are the most widely studied subclasses in clinical research. As previ-ously mentioned, total HDL can be further divided into HDL2 and HDL3 based on density. This allows researchers to independently assess the predictive value of both HDL2 and HDL3 in relation to CVD to determine which subclass has a stronger association. However, the question remains: which subclass is primarily responsible for the inverse association between HDL and CVR? Is it HDL2 or HDL3? There are three main viewpoints: (1) HDL2 is the strongest predictor of CVR [35-37]; (2) HDL3 is the main determinant of the negative association between HDL and CVR[38-40]; (3) Compared to total HDL-C, neither HDL2 nor HDL3 provides additional predictive value for CVR[41,42]. Below, we sum-marize clinical studies supporting each of these viewpoints.”

  • page 5, line 150-177

(paragraph 1)First, there are studies that support the first opinon.。。。。。。

(paragraph 2)Next, other scientists hold the second opinion.。。。。。。

(paragraph 2)At last, there is the third opinion.。。。。。。

Comments 5: Authors briefly mention ezetimibe which is a very used lipid lowering medication drug, since it is widely used, and authors have mentioned almost all the other classes of lipid lowering medication, perhaps ezetimibe deserves a paragraph of it's own.

Response 5: We sincerely appreciate the valuable comments. We have done our best to search for relevant clinicla studies on the association of HDL subclasses with ezetimibe. Upon screening, we regret to find that there are too few studies available to dedicate a separate paragraph to ezetimibe.

Comments 6: Not only that but the authors do not mention other lipid modifying medication such as omega fatty acids or bile acid sequestrants.

Response 6: We think this suggestion is excellent. We searched the PubMed website using the keywords 'omega fatty acids and HDL subclasses' and 'bile acid sequestrants and HDL subclasses'. While there is abundant literature on omega fatty acids and HDL subclasses, we were unable to find literature specifically related to bile acid sequestrants and HDL subclasses. Therefore, we focused our review on summarizing the literature concerning omega fatty acids. Given that omega fatty acids represent a non-pharmacological approach, we included a new section discussing their impact on HDL subclasses alongside the effects of pharmacological treatments. The influence of omega fatty acids on HDL subclasses was comprehensively explored in this revised section.

The revised text is shown below:

“5.2. Omega fatty acids increase HDL size

Omega fatty acids can inhibit the synthesis of lipids and lipoproteins in the liver, thereby reducing plasma cholesterol and TG, which is beneficial for maintaining car-diovascular health. Similar to exercise training, multiple studies have found that in-creasing dietary omega fatty acids is significantly associated with favorable changes in HDL subclasses distribution. For instance, Wooten et al. investigated the effect of short-term omega fatty acids supplementation on HDL subclasses in sedentary men. They found that omega fatty acids caused a shift in HDL subclasses from smaller HDL3a and HDL3b to larger HDL2b and HDL2a[101]. In healthy young adult twins, Bogl et al. also found that omega fatty acids increased HDL2b and decreased HDL3a and HDL3b[102]. Grytten et al. separated small/medium/large HDL using NMR and observed that large HDL incresed and small HDL decresed after omega fatty acids supplementation[103]. Moosavi reported that, compared to the placebo group, the content of large HDL sig-nificantly increased and small HDL significantly decreased in the omega fatty acids group[104]. The reason for this phenomenon is a decrease in CETP activity, which reduces the exchange of cholesterol from HDL to very low-density lipoprotein (VLDL) and LDL, consequently increasing HDL subclasses with large size[105]. The increase in large HDL subclasses suggests potential cardioprotective effects of omega fatty acids, consistent with the effects of exercise training. Therefore, in addition to hypolipidemic drugs, a com-bination of exercise and dietary omega fatty acids should be considered for more effective prevention and treatment of CVD.” (page 11, line 366-386).

Comments 7: The paragraph regarding niacin is rather short and lacking in detail.

Response 7: Thank you for your helpful suggestion. In the revised manuscript, we have incorporated additional details regarding niacin as outlined below.

“Specifically, in patients with primary hypercholesterolemia, Morgan et al. found that extended-release niacin (ERN) increased large HDL subclasses (HDL2a, HDL2b) but did not affect small HDL (HDL3a, HDL3b, HDL 3c)[80]. Similarly, Franceschini and col-leagues observed no change in the number of small HDL after ERN intervention, while the large HDL significantly increased[84]. Besides, Lamon-Fava et al. also concluded that ERN promoted the maturation of HDL into larger particles[82]. Be partial different from Morgan and Franceschini's findings, Kuvin[81] and Jafri[83] et al. found that ERN not only significantly increased the large HDL but also significantly reduced the small HDL. ” (page 10, line 300-308)

Comments 8: The authors fail to mention the potential adverse effects of cetp inhibitors and controversies regarding this class of drugs.

Response 8: Thank you for your helpful suggestion. We have checked the literature carefully and added more references on CETP inhibitors into the 4.2.3 Part. The potential adverse effects of CETP inhibitors and controversies regarding this class of drugs have been discussed in the revised manuscript.

The revised text is shown below:

“Although CETP inhibitors have been expected as a potential new way to preventing CVD, large clinical trials have showed disappointing results. Because of off-target effects, torcetrapib activated the renin angiotensin aldosterone system, leading to side effects such as high aldosterone and cortisol levels and electrolyte abnormalities[91]. Dalcetrapib and evacetrapib did not exhibit torcetrapib-like toxicity but were terminated early due to clinical futility[92,93]. Anacetrapib was associated with a significant 9% reduction in cardiovascular events. However, after stopping administration, prolonged lipid effects were observed, leading to adipose tissue accumulation in subjects. Thus, anacetrapib was not approved by regulators finally[94]. New CETP inhibitors are still being evaluated in large clinical trials. If there is still little cardiovascular benefit after a significant increase of HDL-C levels, it is crucial to carefully consider how to increase HDL subclasses that are truly cardioprotective.” (page 10, line 327-338)

Comments 9: Line 289 authors state "As a natural biological nanocarrier, HDL can transport specific contrast agents and drugs to target tissues, improving diagnosis and treatment efficiency" but they should revise this statement and be clearer with what practical applications does HDL have in such situations.

Response 9: Thank you for your helpful suggestion. We have revised this statement and added practical applications of HDL as a transport vehicle in the revised manuscript, as shown below.

HDL is a natural nanoparticle capable of transporting cholesterol in mammalian circulation. Due to their nanoscale size, long half-life, and abundance of lipids and proteins, HDL has a strong potential as a delivery vehicle. For example, Lin et al. used natural HDL isolated from human plasma to encapsulate 10-hydroxycamptothecin (HCPT)[109]. To overcome the blood-brain barrier and blood-brain tumor barrier, they also added dual modifications with T7 and dA7R peptide ligands to the HDL. Ultimately, this natural HCPT-loaded T7/dA7R-HDL was able to target glioma cells and demonstrated excellent anti-glioma efficacy. Besides, David et al. replaced the hydrophobic core of HDL with inorganic nanocrystals, creating novel HDL-based contrast agents successfully used in computed tomography, magnetic resonance, and fluorescence imaging[110]. However, purifying HDL from human plasma carries the risk of infection and is quite cumbersome. To prepare enough HDL in high quality for in vivo therapeutic use, researchers have used industrial methods to synthesize recombinant high-density lipoprotein (rHDL), such as sodium cholate dialysis method, sonication method, and thermal cycling method[111]. rHDL offers advantages of high stability, long storage time, ease of synthesis, and con-sistent quality from batch to batch. Studies mainly divide rHDL into two subclasses: discoidal rHDL (d-rHDL) and spherical-rHDL (s-rHDL), each with its own characteristics. (page 12, line 403-419)

Round 2

Reviewer 1 Report

Comments and Suggestions for Authors

The revised manuscript was finely corrected.

Comments on the Quality of English Language

No comment

Author Response

Comments 1: The revised manuscript was finely corrected.

Response 1: We greatly appreciate your understanding of the merits of our revised manuscript and the positive comments. Firstly, we apologize for the previous language issues. We have carefully reviewed and improved the language and readability of the manuscript and have also involved native English speakers for further language correction. Finally, we sincerely appreciate your diligent efforts and hope that the revised manuscript can satisfy the requirements for publication.

Reviewer 3 Report

Comments and Suggestions for Authors

The authors have mainly responded to my commentary and have adapted the manuscript accordingly in most regards.

The modified title states that this is "A review", I would go further to mention what sort of review this is (example: a narrative review) along with further polishing the manuscript. 

Comments on the Quality of English Language

Minor English revisions needed

Author Response

Comments 1: The authors have mainly responded to my commentary and have adapted the manuscript accordingly in most regards.

Response 1: We greatly appreciate your understanding of the merits of our revised manuscript and the positive comments. 

Comments 2: The modified title states that this is "A review", I would go further to mention what sort of review this is (example: a narrative review) along with further polishing the manuscript. 

Response 2: Thanks for your valuable advice. We have changed the title to 'HDL Subclasses and Their Role in Prevention and Treatment of Cardiovascular Disease: A Narrative Review'.